

# Sunlight, Clouds, Sea Ice and Albedo: The Umbrella Versus the Blanket

Donald K. Perovich

[1]Thayer School of Engineering, Dartmouth College, Hanover,03755, United States of America

*Correspondence to*: Don Perovich (donald.k.perovich@dartmouth.edu )

**Abstract.** The surface radiation budget plays a central role in summer ice melt and is governed by clouds and surface albedo. I calculated the net radiation flux for a range of albedos under sunny and cloudy skies and determined the break-even value, where the net radiation is the same for cloudy and sunny skies. Break-even albedos range from 0.30 in September to 0.58 in July. For snow covered or bare ice, sunny skies always result in less radiative heat input. In contrast, leads always have, and

ponds usually have, more radiative input under sunny skies than cloudy skies. Snow covered ice has a net radiation flux that is negative or near zero under sunny skies, resulting in radiative cooling. Areally averaged albedos for sea ice in July result in a smaller net radiation flux under cloudy skies. For the other four months, the net radiation is smaller under sunny skies.

## 1. Introduction

The Arctic sea ice cover has undergone a major decline in recent decades. There has been a reduction in ice extent (Meier et

al., 2014; Parkinson and DiGirolamo, 2016), ice thickness (Kwok and Rothrock, 2009; Laxon et al., 2013; Lindsay and Schweiger, 2015), and a shift towards younger ice (Maslanik et al., 2011). This younger, thinner ice cover is more vulnerable to forcing from the atmosphere and ocean. Understanding the feedbacks and forcing driving these changes is critical to improving predictions of ice extent.

Longwave and shortwave radiation are primary drivers in the surface heat budget during summer melt (Persson et al.,

2002). The surface radiative balance consists of contributions from incoming shortwave radiation, reflected shortwave radiation, incoming longwave radiation, and outgoing longwave radiation. Clouds have a major impact on both the incoming longwave and shortwave radiative fluxes. In the winter, the impact is straightforward: clouds warm the surface. The situation in the summer is more complex with clouds playing two opposing roles. They act as an umbrella, cooling the surface by reducing the incoming shortwave radiation. They also act as a blanket, warming the surface by increasing the incoming

longwave radiation. Which effect dominates depends on the type of clouds and the albedo of the surface. Previous work by Interieri et al. (2002) used data from the Surface Heat Budget of the Arctic Ocean (SHEBA) field campaign (Perovich et al., 1999; Uttal et al., 2002) to show that for most of the year clouds acted to warm the surface, but there was a period in summer when clouds cooled the surface.



Here we further explore the impact of clouds on the radiative balance from the perspective of ice surface conditions and albedo during SHEBA. As the melt season progresses from May through September, the ice surface conditions and the albedo change, influencing the net radiation balance. Figure 1 shows the evolution of the ice cover and the albedo from May through September including pre-melt, the melt season, and fall freezeup. Prior to melt the surface is a mix of snow covered ice and open water. During melt, it is a complex, evolving matrix of bare ice, melt ponds, and open water. The albedos of these surfaces range from 0.07 for open water (Pegau and Paulson, 2001) to 0.85 for snow covered ice (Perovich et al., 2002a). Significant temporal evolution of the albedo occurs during this period, along with a great increase in spatial variability. We will explore the impact of these changes in surface albedo on the net radiative forcing for sunny and cloudy sky conditions.

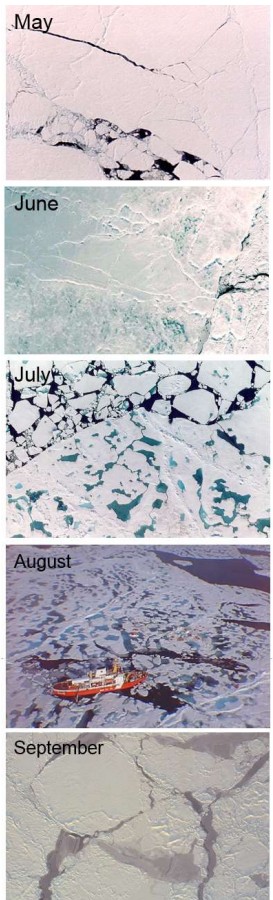

**Figure 1.** Aerial photographs showing the evolution of ice conditions from pre-melt in May through melt in June, July, and August, and freezeup in September.




## 2. Methods

I used data from the SHEBA field campaign (Uttal et al., 2002) to compare the net radiative forcing for sunny and cloudy skies for different surface conditions and albedos. The SHEBA dataset contains a complete observational record of the radiative fluxes (Persson et al., 2002), cloud conditions (Interieri et al., 2002), and ice conditions and albedo (Perovich et al.,

2002a, b). The focus of the prior work and this study, was on radiative fluxes and the relatively small turbulent fluxes were not considered (Persson et al., 2002; Interieri et al., 2002). Using these datasets, I selected pairs of 24-hour-long periods of sunny and cloudy conditions for each month from May through September, with the pairs as close together in time as possible. It was challenging to get a 24 hour, complete solar cycle of uniform sky conditions, particularly for sunny skies in July and August when clouds are pervasive.

The selection of cloudy and sunny pairs from the complete summer SHEBA database was a three-step process. First values of the incoming longwave and shortwave radiation (Persson et al., 2002) were examined. Periods of relatively small incoming shortwave and large incoming longwave were identified as potential cloudy periods, while large incoming shortwave and small incoming longwave were possible sunny periods. The cloud properties of these periods were then examined using the SHEBA cloud data browser of the NOAA Earth Systems Research Laboratory

(https://www.esrl.noaa.gov/psd/arctic/sheba/browser/index.html). This database has a complete record of radiometer and lidar cloud retrievals from SHEBA. The final step was check the qualitative description of sky conditions recorded in logs by observers in the field during the experiment. Table 1 lists the selected periods.

For each of the selected cases the net radiation flux was calculated using

$$F_{net} = (1 - \alpha)S + L^{\downarrow} - L^{\uparrow} \ , \quad (1)$$

where S is the incoming shortwave radiation, $L^{\downarrow}$ is the incoming longwave radiation, $L^{\uparrow}$ is the outgoing longwave radiation, and $\alpha$ is the albedo of the surface. The sign convention is positive to the surface (incoming). Values observed during the SHEBA experiment (Uttal et al., 2002) provide the radiation fluxes (Persson et al., 2002) and the albedos (Perovich et al., 2002a).

To explore the impact of albedo we calculated a zero net albedo ($\alpha_o$) and a break-even albedo ($\alpha_e$). The zero net albedo is

defined as the albedo that for a given radiative forcing results in a value of $F_{net} = 0$.

$$\alpha_o = \frac{[S + L^{\downarrow} - L^{\uparrow}]}{S}$$

For albedos greater than $\alpha_o$, there is net radiative cooling. There are radiative forcings where $\alpha_o$ does not falls within the allowable albedo range of 0 to 1. For example, in early spring, the incoming shortwave is small, skies are often sunny and the outgoing longwave is larger than the incoming longwave, giving a negative value of $\alpha_o$. This means the net radiation is

negative regardless of albedo. In contrast, if the incoming longwave is greater that outgoing, then $\alpha_o$ will always be greater than one and the net radiation will always be positive regardless of the albedo.



**Table 1.** Summary of monthly sunny / cloudy pairs including longwave and shortwave fluxes (W m$^{-2}$), break-even albedos, and zero net albedos. $L^{\downarrow}$ and $L^{\uparrow}$ are the incoming and outgoing longwave radiation, and S is the incoming shortwave.

| Start | End | Sky conditions | Break-even albedo | Zero net albedo | $L^{\downarrow}$ | $L^{\uparrow}$ | S |
|---|---|---|---|---|---|---|---|
| *May* | | | 0.40 | | | | |
| 5/24/98 0:00 | 5/24/98 24:00 | Clear | | 0.80 | 205 | 275 | 345 |
| 5/29/98 0:00 | 5/29/98 24:00 | Cloudy | | na | 320 | 314 | 219 |
| | | | | | | | |
| *June* | | | 0.54 | | | | |
| 6/15/98 0:00 | 6/15/98 24:00 | Clear | | 0.82 | 231 | 301 | 389 |
| 6/16/98 16:00 | 6/17/98 16:00 | Cloudy | | na | 315 | 308 | 218 |
| | | | | | | | |
| *July* | | | 0.58 | | | | |
| 7/25/98 13:00 | 7/26/98 13:00 | Clear | | 0.83 | 265 | 313 | 284 |
| 7/22/98 0:00 | 7/22/98 24:00 | Cloudy | | na | 320 | 316 | 161 |
| | | | | | | | |
| *August* | | | 0.38 | | | | |
| 8/27/98 10:00 | 8/28/98 10:00 | Clear | | 0.73 | 258 | 294 | 132 |
| 8/26/98 3:00 | 8/27/98 3:00 | Cloudy | | 0.95 | 303 | 308 | 81 |
| | | | | | | | |
| *September* | | | 0.30 | | | | |
| 9/4/98 20:00 | 9/5/98 20:00 | Clear | | 0.63 | 267 | 293 | 71 |
| 9/2/98 10:00 | 9/3/98 10:00 | Cloudy | | 0.81 | 294 | 303 | 47 |

The break-even albedo is the albedo where the net radiation for cloudy skies (subscript *c*) is the same as sunny skies

5 (subscript *s*).

$$F_{netc} = F_{nets}$$

$$(1 - \alpha_e)S_s + L_s^{\downarrow} - L_s^{\uparrow} = (1 - \alpha_e)S_c + L_c^{\downarrow} - L_c^{\uparrow}$$

Solving for $\alpha_e$ gives

$$\alpha_e = 1 - \left[ \frac{\left| L_c^{\downarrow} - L_s^{\downarrow} - L_c^{\uparrow} + L_s^{\uparrow} \right|}{S_s - S_c} \right]$$

10 For albedos greater than the break-even albedo, sunny skies have a smaller net radiation than cloudy skies. If the albedo is less than the break-even value, the net radiation is smaller for cloudy skies.





### 3. Results

Figure 2 shows one of the sunny (June 15) and cloudy (June 17) sky data pairs. There is very little difference in the outgoing longwave as the surface was near 0°C in both cases. The incoming longwave is smaller at all times for the sunny skies, with a sunny sky daily average of 230 Wm$^{-2}$ compared to 315 Wm$^{-2}$ for cloudy skies. The sunny sky incoming shortwave

radiation is 1.5 to 2 times the cloudy sky values during the day, with a daily average incoming shortwave of 388 Wm$^{-2}$ for sunny skies and 218 Wm$^{-2}$ for cloudy skies.

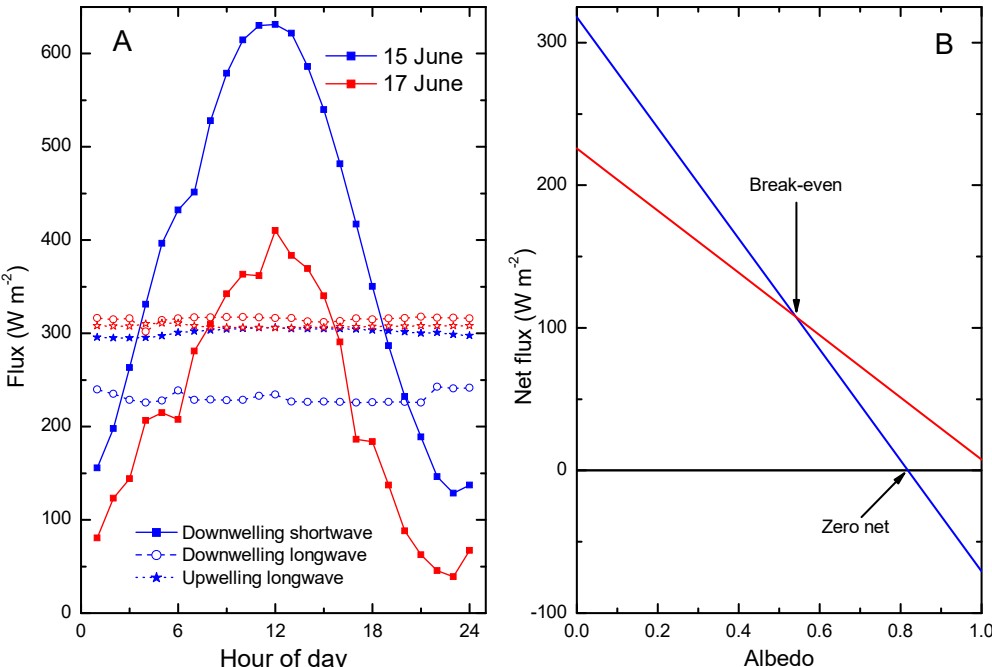

**Figure 2.** Results from sunny (15 June) and cloudy (17 June) skies: A) Hourly values of incoming longwave and shortwave radiation and outgoing longwave. B) The net radiation as a function of albedo with the break-even and zero net albedos

indicated.

The net radiation flux as a function of albedo (Equation 1) is plotted in the Figure 2B. The slope of the sunny sky line is steeper than the cloudy sky line because the incoming shortwave is larger. The sunny and cloudy lines intersect at a break-even albedo of 0.54. For albedos larger than 0.54 the net radiation flux is smaller for sunny skies than cloudy skies. For

sunny skies, the zero net albedo is 0.82 and larger albedos result in a net radiative cooling of the surface. This implies that



dry snow, with an albedo of about 0.85, would experience radiative cooling under the sunny sky conditions. For cloudy skies, the net radiation flux is always positive, regardless of the albedo.

Table 1 summarizes results for the sunny/cloudy pairs selected for each month from May through September. The dates, sky conditions, daily averaged shortwave and longwave radiation, and zero net albedos are reported for each day, along with
the break-even albedo for each month. Break-even albedos range from a low of 0.30 in September, when the incoming shortwave is small, to a peak value of 0.58 in July. Under sunny skies, there is a physically possible zero net albedo for every month. In May, June, and July values are 0.80 to 0.83 implying that snow covered ice could experience a slight radiative cooling. In August and September, when the incoming shortwave radiation is smaller, the sunny sky zero net albedos are 0.73 and 0.63 respectively. For cloudy skies, only September has a physically possible value (0.81) for the zero net albedo.

The sunny / cloudy sky impact on different ice types is examined in Figure 3. The sunny minus cloudy average daily net radiation for the five monthly pairs is plotted as a function of albedo. The slope of the line depends primarily on the difference between the sunny and cloudy incoming shortwave radiation. The steepest slope is in June when the incoming shortwave is the largest and the shallowest is in September when the incoming shortwave radiation is smallest. Also plotted are the lines denoting typical albedos (Pegau and Paulson, 2001; Perovich et al., 2002a) for leads ($\alpha_w$= 0.07), bare ice ($\alpha_i$ =
0.65), and snow ($\alpha_s$ = 0.85) plus a shaded box showing the range of pond albedos ($\alpha_p$ = 0.15 to 0.40). Under cloudy skies, the net radiative flux is always less for leads and almost always less for ponds. For light blue ponds in May, August, and September there is little difference in net radiation between cloudy and sunny skies. In contrast, the net radiative flux is always less under sunny skies for bare ice and snow and is substantially less in May and June.

Break-even albedos for the five monthly pairs are plotted in the insert. Values increase from May to June, reach a
maximum in July, and then decrease in August and September. The July maximum in break-even albedo is largely due to a reduced contrast between sunny and cloudy incoming longwave radiation compared to June values.

On the aggregate scale the ice cover is an ensemble of surface conditions including snow covered ice, bare ice, ponds, and leads, with significant spatial variability and temporal evolution. The impact of sunny and cloudy skies on the net radiation flux for the aggregate scale ice cover depends on the aggregate scale albedo $\alpha_g$, which is a function of the composition of the
ice cover:

$$\alpha_g = \alpha_{si} A_{si} + \alpha_y A_y + \alpha_p A_p + \alpha_w A_w$$

The area fractions of snow covered and bare ice ($A_{si}$), ponded ice ($A_p$), leads ($A_w$), and young ice ($A_y$) (Table 2) were determined from aerial photographs from May through September during the SHEBA experiment (Perovich et al., 2002b) The albedos of these surface types were measured throughout the summer (Perovich et al., 2002a).

Aggregate scale albedos for SHEBA range from a minimum of 0.50 in July to a maximum of 0.82 in May. The net radiation flux on the aggregate scale is smaller for sunny skies than cloudy skies in May, June, August, and September. Only July has a net radiation flux greater for sunny skies than cloudy. The 10% higher net radiation for sunny skies is due to the small value of the aggregate-scale albedo and to the reduced difference between sunny and cloudy sky incoming longwave





radiation. July is also the period of the largest net radiative flux for both sunny and cloudy skies. The net radiative flux is always positive for cloudy skies. For the May and September cases, the aggregate scale net radiative flux is negative under sunny skies. Thus, sunny skies can delay the onset of melt in May and facilitate the onset of freezeup in September.

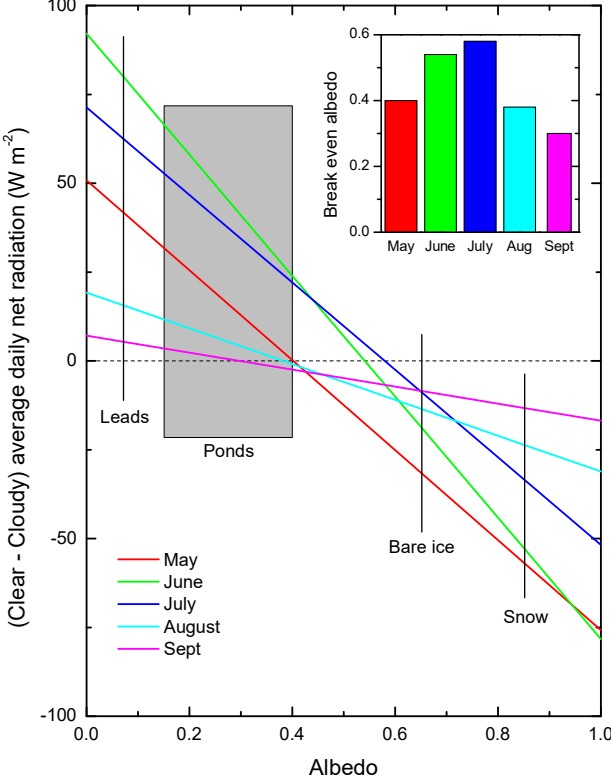

**Figure 3**. Sunny minus cloudy average daily net radiation as a function of albedo. The albedos of leads, ponds, bare ice, and snow are plotted for reference. The insert shows the break-even albedo for each monthly sunny/cloudy pair.

## 3. Discussion

These results indicate that for the aggregate scale ice cover during SHEBA, the net radiation flux is smaller under sunny
10   skies compared to cloudy for every summer month except July. These results are for the ice around a ship in the Beaufort Sea drifting northward from 76° N in May 1998 to 79° N in September 1998. The advantage of using this case is that there are extensive data on the state of the ice cover, cloud conditions, and radiative fluxes. It is possible, with some simplifying assumptions, to extrapolate the results of this study. For example, assuming the cloud properties remain the same and





changing the latitude would change the incoming shortwave radiation and change the break-even albedo. Moving to higher latitudes decreases both, making sunny skies more beneficial to maintaining the ice cover. Changes in the timing of the onset dates of melt and freezeup, as well as changes in the age of the ice, also influence the evolution of albedo and consequently the sunny / cloudy sky balance. Net radiation is less under sunny skies for changes that result in more snow covered or bare

ice. Changes increasing ponds and leads favor cloudy skies for smaller net radiation.

We can explore the potential impact of changing ice conditions on the aggregate scale by comparing SHEBA in 1998 to the same region in 2007, which had a record summer minimum ice extent, with significant ice loss in the Beaufort Sea. The 2007 ice concentration (NSIDC-5001) and pond fraction (Rosel et al., 2012) were determined from satellite data at the same locations as the 1998 SHEBA cases in Table 1. The area fractions and areally averaged albedos for the two years are

summarized in Table 2 and plotted in Figure 4.

**Table 2.** Monthly aggregate-scale ice cover composition and albedo from May through September in 1998 and in 2007.

| Date | Net radiative flux (W m-2) Sunny | Cloudy | Areally averaged albedo | Break even albedo | Snow / ice fraction | Snow / ice albedo | Lead fraction | Lead albedo | Pond fraction | Pond albedo | Young ice fraction | Young ice albedo |
|---|---|---|---|---|---|---|---|---|---|---|---|---|
| *1998* | | | | | | | | | | | | |
| May | -7.1 | 45.6 | 0.82 | 0.40 | 0.97 | 0.84 | 0.03 | 0.07 | 0 | | 0 | |
| June | 48.6 | 74.6 | 0.69 | 0.54 | 0.93 | 0.73 | 0.03 | 0.07 | 0.04 | 0.35 | 0 | |
| July | 93.9 | 84.2 | 0.50 | 0.58 | 0.73 | 0.63 | 0.05 | 0.07 | 0.22 | 0.17 | 0 | |
| August | 17.9 | 28.6 | 0.59 | 0.38 | 0.80 | 0.72 | 0.18 | 0.07 | 0.02 | 0.40 | 0 | |
| September | -0.6 | 7.8 | 0.65 | 0.30 | 0.72 | 0.84 | 0.13 | 0.07 | 0 | | 0.15 | 0.20 |
| | | | | | | | | | | | | |
| *2007* | | | | | | | | | | | | |
| May | -15.0 | 40.5 | 0.84 | 0.40 | 1.00 | 0.84 | 0.00 | 0.07 | 0.00 | | 0 | |
| June | 131.7 | 121.3 | 0.48 | 0.54 | 0.50 | 0.73 | 0.21 | 0.07 | 0.29 | 0.35 | 0 | |
| July | 133.9 | 106.9 | 0.36 | 0.58 | 0.48 | 0.63 | 0.31 | 0.07 | 0.21 | 0.17 | 0 | |
| August | 54.0 | 50.9 | 0.32 | 0.38 | 0.31 | 0.72 | 0.54 | 0.07 | 0.15 | 0.40 | 0 | |
| September | 28.6 | 27.2 | 0.24 | 0.30 | 0.14 | 0.84 | 0.80 | 0.07 | 0.05 | | 0 | 0.20 |

To focus on the impact of changing ice conditions, the observed radiative fluxes from SHEBA were used in the 2007

analysis. This means that the break-even albedo is the same in 2007 as in 1998. The 1998 observed albedos for snow, bare ice, ponds, and leads were also used in 2007. The focus is on the impact of changes in the state of the ice cover. Ice conditions in the SHEBA region were markedly different in 2007 compared to 1998. In 2007, there was much more open water from June through September were much smaller and much less bare ice. Pond fractions were much larger in June and August of 2007 (Figure 4, Table 2). In 1998 ponds were freezing in August greatly reducing the pond fraction and young ice





was forming in September. Freezeup was later in 2007, ponds were not freezing in August and young ice was not forming in early September.

Aggregate scale albedos in May for the two years are comparable (0.82 and 0.84). However, from June through September aggregate scale albedos were substantially smaller in 2007 than in 1998 (Table 2). More leads and more melt ponds in 2007

5  resulted in a smaller aggregate scale albedo and a considerable increase in the net radiative balance. The decrease in albedo also changed the sunny / cloudy dynamic. For the ice conditions observed in 2007, with more ponds and leads, the albedo was less than the break-even albedo in every month except May. Thus, cloudy skies were more favorable to maintaining the ice cover from June through September.

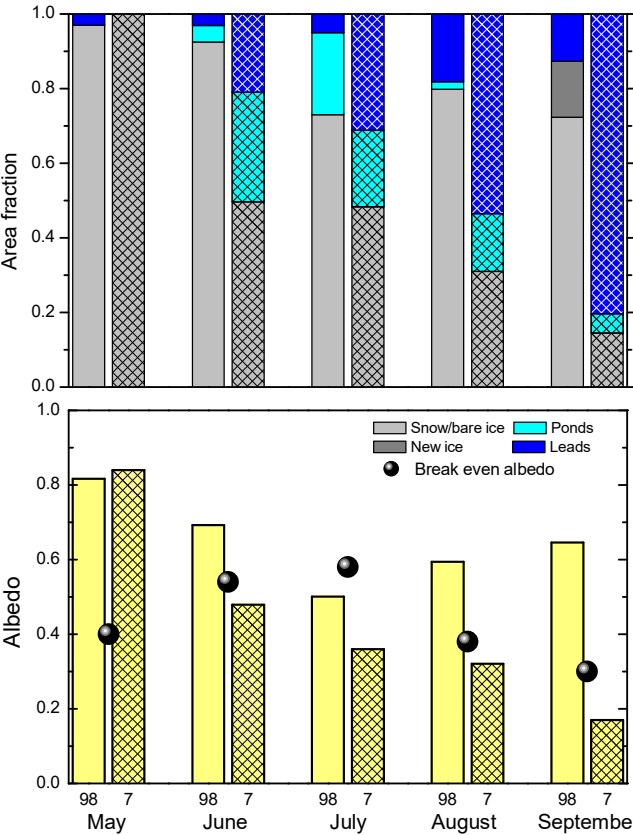

10  **Figure 4.** Top panel: Area fractions of snow/bare ice, ponds, leads, and new ice in 1998 and 2007. Bottom panel: Aggregate scale albedos for 1998 and 2007 along with break-even albedos. Solid bars show 1998 results and hatched bars show 2007.



Ice conditions will continue to dictate whether sunny or cloudy skies result in smaller net radiative fluxes. Observed changes to the summer ice cover in recent decades show less ice, younger ice, and early melt onset all reduce the aggregate scale albedo, increasing the radiative flux under both sunny and cloudy skies. However, the increase is greater under sunny skies. The SHEBA data showed that sunny skies in 1998 often provided a modest respite from surface melt. This may not be

true in the future.

## 4. Conclusions

Using field observations from the SHEBA program, I selected pairs of sunny and cloudy days for each month from May through September and calculated the net radiation flux for various surface conditions and albedos. To explore the impact of albedo, I calculated a break-even albedo, for which the net radiation for cloudy skies is the same as for sunny skies. For

albedos larger than the break-even value, the net radiation flux is smaller under sunny skies than cloudy skies. Break-even albedos range from 0.30 in September to 0.58 in July. For snow covered or bare ice, sunny skies always result in less radiative heat input. In contrast, leads always have, and ponds usually have, more radiative heat input under sunny skies than cloudy skies. Under sunny skies, snow covered ice has a net radiation flux that is negative or near zero, resulting in radiative cooling.

Aggregate scale albedos calculated using results from SHEBA show that sunny skies usually result in reduced radiative heat input. For May, June, August, and September, the areally averaged albedo is greater than the break-even albedo, favoring sunny skies. For the May and September cases, the areally averaged net radiation flux is even negative under sunny skies. It is only for the July case that the areally averaged albedo of 0.50 is less than the break-even albedo, resulting in a smaller net radiation flux under cloudy skies than under sunny skies.

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
