# Peer review of "Sunlight, Clouds, Sea Ice, Albedo and the Radiative Budget: The Umbrella Versus the Blanket"

_The Cryosphere, 2018_

## Referee Comment (RC1) · Anonymous Referee #1 · 3 Apr 2018

Review of the manuscript entitled 'Sunlight, Clouds, Sea Ice and Albedo: The Umbrella Versus the Blanket' by Donald K. Perovich

This is a well-written and concise manuscript that in a very simple and straightforward manner investigates the coupled effects of cloud radiative feedback and ice-albedo feedback on the Arctic Ocean surface radiation budget. Its strength is precisely the simplification that are made that help promote an understanding of the coupled sea ice - atmosphere system, and stimulates ideas for further research. However, these simplifications might result in a fundamental flaw in the interpretation of the results, and it is not clear how to directly translate these results into real-world situations which are dominated by variability. I think the basic problem is that conditions are considered constant, and averaged, over a 24h period, while in fact there are significant diurnal varia-

tions, especially in the incoming shortwave radiation (see Fig. 2A), but also cloudiness. The interpretation of the 24h-averages is that sunny skies cause less melting of sea ice surfaces than cloudy skies. However, during a clear sky 24h period, the shortwave radiation would promote positive net radiation balance (surface warming/melting) during daytime, and a negative balance (cooling/refreezing) during nighttime. Such day and night differences would have repercussion on what the actual break-even and zero-net albedos would be. The largest melting would appear to occur during situations when the daytime was clear sky and the nighttime was cloud covered. I don't think it would be so much more work to expand this discussion to include a basic consideration of diurnal effects, and it looks to me that the used dataset would support such analysis without too much added complexity. Furthermore, and I do not think it is necessarily needed to illustrate the point of the manuscript, is to expand it towards a more rigorous statistical analysis by incorporating more surface radiation balance datasets from various locations and time periods. There are quite a few such datasets.

Some specific comments:

Title: I would suggest including 'radiative budget' in some way in the title.

Abstract: This is a very short and 'stoic' abstract. If allowed by the journal, some more information and explanation could be included to make the abstract more self-explanatory.

page 1, line 25. In addition to cloud and albedo, there are also effects from the solar zenith angle. See e.g. Minnett (1999, doi:10.1175/1520-0442-12.1.147), which could be included as a citations in the manuscript.

Table 1. Instead of 'na', consider including the positive or negative 'net zero albedo' values.

page 5, line 15. 'Net radiative cooling'. It would be good to clarify here and elsewhere that this is for a daily average.

page 6, line 9. 'Five monthly pairs' is a very small dataset to justify Arctic wide conclusions.

page 6, line 22. From 'On the aggregate scale...' to the end of results: Consider moving this into the Discussion section.

page 7, line 3: 'Thus, sunny skies can delay the onset of melt in May and facilitate the onset of freezeup in September.' This general statement might need re-evaluation of diurnal effects are considered.

---

## Referee Comment (RC2) · Anonymous Referee #2 · 4 Apr 2018

This manuscript describes a set of calculations carried out to assess the magnitudes of surface radiative forcing for a sea ice cover during summer. The calculations present relative differences between cloudy and clear conditions for the 5 months typically spanning melt season in the Arctic. The data employed are taken from the SHEBA project.

The topic is interesting and relevant for TC readership. The presentation is clear, original, and insightful. I agree that sweeping simplifications were made, but for the purpose of illustrating how these concepts apply to the real world Arctic, they seem justified and appropriate. This is a simple, yet clear, illustration of how radiative forcing responds to clouds and surface albedo for a sea ice cover. It is true that the analysis is done exclusively for individual "snapshot" samples in time, but the conclusions are informative

and instructional.

Minor comments: Abstract line 6: First sentence should state the domain you are considering—"The surface radiation budget of the Arctic Ocean plays..."

Abstract line 12: "other four months" is not clear, could say "other four months of the melt season" or explicitly spell them out "For May, June, August, and September, the net..."

p. 1 Line 18: Just improving prediction of ice extent? What about also improving prediction of ice thickness (specifically surface melt)?

p. 3 line 10: "First," p. 3 line 16: "to check" p.3 line 25: : "...defined as the albedo that, for a given radiative forcing, results..." p.3 line 27: "fall" p. 3 line 30: "...is greater than outgoing..."

Fig. 2: The legends would be easier to decode if they were more consistent with the terminology used in the text. I recommend using "sunny" and "cloudy" for the color legend (it's totally clear in the text which date is which condition, but it's not intuitive in the figure and someone looking at the figure won't really care what the date is, they just care about the sky conditions). Also, the line type legend would be clearer if stated as "incoming" and "outgoing", as used in the caption and the text.

p. 6 line 15- 16: This sentence is a bit unclear. Does it mean "Under cloudy skies the net radiative flux is always less than for clear skies for leads and almost always less for ponds."?

p. 8 line 18: can't tell if the end of this sentence is a copy and paste accident or whether there are commas missing, but it needs to be rewritten

p. 8 line 19: "...freezing in August, greatly reducing the pond fraction, and young ice..."

p. 10 line 15 - 16: Does this mean that one should expect to see significant differences
in surface melt between cloudy and sunny conditions? This may merit a reference to Perovich et al. 2003, where this idea was posed.

p. 10 line 17: Not sure what "favoring" means here? How about "For M, J, A, and S, the . . . albedo is greater than the break-even albedo, suggesting that sunny skies promote less surface melt"?

And, finally, a question: Does this analysis suggest that a cloudy period is required (or even just hugely beneficial) to the initiation of melt in the early summer? If so, this might be a nice conclusion.
* * *

---

## Referee Comment (RC3) · Anonymous Referee #3 · 16 Apr 2018

The manuscript presents an insightful analysis on the role of sunny and cloudy skies in the surface radiation budget as the albedo seasonally evolves. The effect of changing sea ice conditions on the net radiative flux was investigated by comparing sea ice conditions in the Beaufort Sea in 1998 and 2007. The main findings demonstrate that sunny skies had a lower net radiation flux in May and September, while cloudy skies had a lower net radiative flux in June-August in 1998. For 2007, cloudy conditions had a smaller net radiative flux than sunny conditions in June-September due to increased melt pond and open water coverage. The results are informative to the sea ice and broader communities, and hint at the changing sensitivity of the sea ice cover to atmospheric conditions and its feedback on the surface radiation budget in a changing Arctic system.

[Figure]

The manuscript is well-organized, the methodology and assumptions clearly described and justified, and the conclusions well-supported by the results. Please find suggestions below that I hope the author will find useful:

Pg. 1, Abstract: Similar to the comment for the conclusion, it would be helpful to include 1-2 sentences describing how the results relate to the broader picture of sea ice-atmosphere interactions in a changing Arctic.

Pg. 1, Line 12: Which four months?

Pg. 1, Line 25: Typo here and throughout the text for "Intrieri."

Pg. 3, Lines 8-9: "It was challenging..." How sensitive are the zero net and break-even albedo values to 24 hours of slightly vs. highly variable conditions?

Pg. 3, Lines 10-16: It would be useful to include the cloud cover and type if that information is available.

Pg. 3, Line 16: Typo "to check."

Pg. 3, Line 19: Please enumerate all equations.

Pg. 3, Line 27: Typo in "fall."

Pg. 4, Table 1: Although it's already described in the text, it would be helpful to include a brief sentence in the table caption explaining "na."

Pg. 8, Line 17-18: Please clarify "In 2007,... bare ice."

Pg. 10, Conclusions: It would be helpful to expand on the implications of the results here. How do they relate to the big picture? What was learned by comparing the 1998 and 2007 sea ice conditions?

---

## Author Comment (AC1)

Review of the manuscript entitled 'Sunlight, Clouds, Sea Ice and Albedo: The Umbrella Versus the Blanket' by Donald K. Perovich

This is a well-written and concise manuscript that in a very simple and straightforward manner investigates the coupled effects of cloud radiative feedback and ice-albedo feedback on the Arctic Ocean surface radiation budget. Its strength is precisely the simplification that are made that help promote an understanding of the coupled sea ice - atmosphere system, and stimulates ideas for further research. However, these simplifications might result in a fundamental flaw in the interpretation of the results, and it is not clear how to directly translate these results into real-world situations which are dominated by variability. I think the basic problem is that conditions are considered constant, and averaged, over a 24h period, while in fact there are significant diurnal variations, especially in the incoming shortwave radiation (see Fig. 2A), but also cloudiness.
The interpretation of the 24h-averages is that sunny skies cause less melting of sea ice surfaces than cloudy skies. However, during a clear sky 24h period, the shortwave radiation would promote positive net radiation balance (surface warming/melting) during daytime, and a negative balance (cooling/refreezing) during nighttime. Such day and night differences would have repercussion on what the actual break-even and zero-net albedos would be. The largest melting would appear to occur during situations when the daytime was clear sky and the nighttime was cloud covered. I don't think it would be so much more work to expand this discussion to include a basic consideration of diurnal effects, and it looks to me that the used dataset would support such analysis without too much added complexity.
*I did perform an hourly calculation of break-even albedo. It was only in September that the sun actually set, but in all cases there were variations in incident sunlight, and break-even albedo from solar midnight to solar noon. For example, in June the break-even albedo ranged from 0.09 at solar midnight to 0.70 at solar noon. I have hourly results for all the cases, but I believe that the daily averages are the most meaningful and so that is what I used.*

Furthermore, and I do not think it is necessarily needed to illustrate the point of the manuscript, is to expand it towards a more rigorous statistical analysis by incorporating more surface radiation balance datasets from various locations and time periods. There are quite a few such datasets.
*I don't think there are many complete datasets in the Arctic ice pack. SHEBA included an entire year with detailed data for clouds, radiative fluxes, albedos, and ice mass balance - a rare combination. I do look forward to analyzing data from the upcoming MOSAiC field campaign.*

Some specific comments:

Title: I would suggest including 'radiative budget' in some way in the title.
*I added Radiative Budget to the title.*

Abstract: This is a very short and 'stoic' abstract. If allowed by the journal, some more information and explanation could be included to make the abstract more selfexplanatory.
*I agree that the abstract is short, but there is a maximum word count of 200.*

page 1, line 25. In addition to cloud and albedo, there are also effects from the solar zenith angle. See e.g. Minnett (1999, doi:10.1175/1520-0442-12.1.147), which could be included as a citations in the manuscript.
*I added this comment and the reference.*

Table 1. Instead of 'na', consider including the positive or negative 'net zero albedo' values.
*I followed Reviewer 3's suggestion and changed "na" to "None" and explained the meaning of "None" in the caption.*

page 5, line 15. 'Net radiative cooling'. It would be good to clarify here and elsewhere that this is for a daily average.
*I added daily average.*

page 6, line 9. 'Five monthly pairs' is a very small dataset to justify Arctic wide conclusions.
*I agree that it is important to be careful about extrapolating. The main point of the*

page 6, line 22. From 'On the aggregate scale: : :' to the end of results: Consider moving this into the Discussion section.
*Good idea. I moved this into the Discussion section.*

page 7, line 3: 'Thus, sunny skies can delay the onset of melt in May and facilitate the onset of freezeup in September.' This general statement might need re-evaluation of diurnal effects are considered.
*The general statement is still true when averaged over a day. It is also true for most hours of the day.*

This manuscript describes a set of calculations carried out to assess the magnitudes of surface radiative forcing for a sea ice cover during summer. The calculations present relative differences between cloudy and clear conditions for the 5 months typically spanning melt season in the Arctic. The data employed are taken from the SHEBA project.

The topic is interesting and relevant for TC readership. The presentation is clear, original, and insightful. I agree that sweeping simplifications were made, but for the purpose of illustrating how these concepts apply to the real world Arctic, they seem justified and appropriate. This is a simple, yet clear, illustration of how radiative forcing responds to clouds and surface albedo for a sea ice cover. It is true that the analysis is done exclusively for individual "snapshot" samples in time, but the conclusions are informative and instructional.

Minor comments: Abstract line 6: First sentence should state the domain you are consideringâ˘Aˇ T"The surface radiation budget of the Arctic Ocean plays: : :"
*Added "of the Arctic Ocean" to the sentence.*

Abstract line 12: "other four months" is not clear, could say "other four months of the melt season" or explicitly spell them out "For May, June, August, and September, the net: : :"
*Spelled out the four months as suggested.*

p. 1 Line 18: Just improving prediction of ice extent? What about also improving prediction of ice thickness (specifically surface melt)?
*Added "thickness, and surface melt" as suggested.*

p. 3 line 10: "First," p. 3 line 16: "to check" p.3 line 25: : ": : :defined as the albedo that, for a given radiative forcing, results: : :" p.3 line 27: "fall" p. 3 line 30: ": : :is greater than outgoing: : :"
*I made all of these changes.*

Fig. 2: The legends would be easier to decode if they were more consistent with the terminology used in the text. I recommend using "sunny" and "cloudy" for the color legend (it's totally clear in the text which date is which condition, but it's not intuitive in the figure and someone looking at the figure won't really care what the date is, they just care about the sky conditions). Also, the line type legend would be clearer if stated as "incoming" and "outgoing", as used in the caption and the text.
*I changed the figure as suggested.*

p. 6 line 15- 16: This sentence is a bit unclear. Does it mean "Under cloudy skies the net radiative flux is always less than for clear skies for leads and almost always less for ponds."?
*I changed this to "is usually less"*

p. 8 line 18: can't tell if the end of this sentence is a copy and paste accident or whether there are commas missing, but it needs to be rewritten
*I did a little rewritting and fixed the punctuation.*

p. 8 line 19: ": : :freezing in August, greatly reducing the pond fraction, and young ice: : :"
*Rewrote this portion to improve clarity.*

p. 10 line 15 - 16: Does this mean that one should expect to see significant differences in surface melt between cloudy and sunny conditions? This may merit a reference to Perovich et al. 2003, where this idea was posed.
*I added a sentence and the Perovich et al., 2003 reference as suggested.*

p. 10 line 17: Not sure what "favoring" means here? How about "For M, J, A, and S, the : : : albedo is greater than the break-even albedo, suggesting that sunny skies promote less surface melt"?
*Changed the text as suggested.*

And, finally, a question: Does this analysis suggest that a cloudy period is required (or even just hugely beneficial) to the initiation of melt in the early summer? If so, this might be a nice conclusion.
*This is an interesting point. There is some speculation at this time that melt onset might be triggered by rain or fog. However, nothing definitive has been determined yet.*

The manuscript presents an insightful analysis on the role of sunny and cloudy skies in the surface radiation budget as the albedo seasonally evolves. The effect of changing sea ice conditions on the net radiative flux was investigated by comparing sea ice conditions in the Beaufort Sea in 1998 and 2007. The main findings demonstrate that sunny skies had a lower net radiation flux in May and September, while cloudy skies had a lower net radiative flux in June-August in 1998. For 2007, cloudy conditions had a smaller net radiative flux than sunny conditions in June-September due to increased melt pond and open water coverage. The results are informative to the sea ice and broader communities, and hint at the changing sensitivity of the sea ice cover to atmospheric conditions and its feedback on the surface radiation budget in a changing Arctic system.

The manuscript is well-organized, the methodology and assumptions clearly described and justified, and the conclusions well-supported by the results. Please find suggestions below that I hope the author will find useful:

Pg. 1, Abstract: Similar to the comment for the conclusion, it would be helpful to include 1-2 sentences describing how the results relate to the broader picture of sea ice-atmosphere interactions in a changing Arctic.
*I added a paragraph to the Conclusions expanding on the implications*

Pg. 1, Line 12: Which four months?
*Spelled out the four months as suggested.*

Pg. 1, Line 25: Typo here and throughout the text for "Intrieri."
*I apologize for the misspelling. I corrected the spelling throughout the text.*

Pg. 3, Lines 8-9: "It was challenging..." How sensitive are the zero net and break-even albedo values to 24 hours of slightly vs. highly variable conditions?

Pg. 3, Lines 10-16: It would be useful to include the cloud cover and type if that information is available.
*There is detailed information from the cloud lidar and radar. However, there is a not a description that is suitable for this paper.*

Pg. 3, Line 16: Typo "to check."
*Typo corrected*

Pg. 3, Line 19: Please enumerate all equations.
*Question to the editor: Should all equations be numbered, or just those that are specifically mentioned.*

Pg. 3, Line 27: Typo in "fall."
*Typo corrected*

Pg. 4, Table 1: Although it's already described in the text, it would be helpful to include a brief sentence in the table caption explaining "na."
*I changed na to "None" and added a sentence to the table caption explaining the meaning of "None."*

Pg. 8, Line 17-18: Please clarify "In 2007,... bare ice."
*Clarified the sentence by minor editing.*

Pg. 10, Conclusions: It would be helpful to expand on the implications of the results here. How do they relate to the big picture? What was learned by comparing the 1998 and 2007 sea ice conditions?
*I added a paragraph to the Conclusions expanding on the implications*